# Mannan-Based Nanodiagnostic Agents for Targeting Sentinel Lymph Nodes and Tumors

**DOI:** 10.3390/molecules26010146

**Published:** 2020-12-31

**Authors:** Markéta Jirátová, Andrea Gálisová, Maria Rabyk, Eva Sticová, Martin Hrubý, Daniel Jirák

**Affiliations:** 1Institute for Clinical and Experimental Medicine (IKEM), 140 21 Prague, Czech Republic; marketa.jiratova@gmail.com (M.J.); a.galisova@gmail.com (A.G.); evsc@ikem.cz (E.S.); 2Department of Physiology, Faculty of Science, Charles University, 128 00 Prague, Czech Republic; 3Institute of Macromolecular Chemistry, Czech Academy of Sciences, 162 00 Prague, Czech Republic; maria.rabyk@gmail.com (M.R.); mhruby@centrum.cz (M.H.); 4Institute of Biophysics and Informatics, 1st Faculty of Medicine, Charles University, 121 08 Prague, Czech Republic

**Keywords:** mannan, SLN, cancer, multimodality imaging, 4T1 cells, MRI

## Abstract

Early detection of metastasis is crucial for successful cancer treatment. Sentinel lymph node (SLN) biopsies are used to detect possible pathways of metastasis spread. We present a unique non-invasive diagnostic alternative to biopsy along with an intraoperative imaging tool for surgery proven on an in vivo animal tumor model. Our approach is based on mannan-based copolymers synergistically targeting: (1) SLNs and macrophage-infiltrated solid tumor areas via the high-affinity DC-SIGN (dendritic cell-specific intercellular adhesion molecule-3-grabbing non-integrin) receptors and (2) tumors via the enhanced permeability and retention (EPR) effect. The polymer conjugates were modified with the imaging probes for visualization with magnetic resonance (MR) and fluorescence imaging, respectively, and with poly(2-methyl-2-oxazoline) (POX) to lower unwanted accumulation in internal organs and to slow down the biodegradation rate. We demonstrated that these polymer conjugates were successfully accumulated in tumors, SLNs and other lymph nodes. Modification with POX resulted in lower accumulation not only in internal organs, but also in lymph nodes and tumors. Importantly, we have shown that mannan-based polymer carriers are non-toxic and, when applied to an in vivo murine cancer model, and offer promising potential as the versatile imaging agents.

## 1. Introduction

It is widely known that cancer is one of the leading causes of death worldwide, and despite the huge progress in cancer treatment over the recent years, many of the mechanisms involved in the complex process of tumor and especially metastasis development are still not fully understood. To address this deficit, efforts are being made to more precisely diagnose and treat metastasis spreading. Axillary lymph node (ALN) status serves as a metastasis prognostic factor. As an alternative to ALN dissection, sentinel lymph node (SLN) biopsy is used as a standard clinical procedure [1,2,3]. SLNs play a key role in metastasis spreading as they form the lymphatic drainage system closest to the tumor and therefore, they are the most probable site of early metastasis [4]. However, SLN biopsy is still an invasive procedure with the possibility of complications. In addition to biopsy, various methods are used for metastasis and SLN detection, including standard blue-dye intraoperative detection and imaging methods such as magnetic resonance imaging (MRI) [5,6,7], ultrasound [8], single-photon emission computed tomography (SPECT), fluorescence, lymphoscintigraphy, and others [9,10]. Each method provides different information, but also comes with limitations, e.g., the low specificity of MRI and the optical signal attenuation. Multimodal imaging is used to overcome these issues. Combining MRI with sensitive and convenient optical imaging provides complementary and precise information on the anatomy as well as the distribution and degradability of contrast agents and drug delivery systems. Reliable imaging methods are crucial for improving both the accuracy of SLN detection and the efficiency of drug delivery systems. To that end, the ultimate goal is to create a specific, selective and non-invasive method of SLN metastasis prediction.

Macrophages, which are heavily present in SLNs, are one of the most important factors in cancer-promoting inflammatory reactions [11] as they are also accumulated in cancer tissue. These tumor-associated macrophages (TAMs) affect inflammation of the stroma among other effects [12]. Macrophages and dendritic cells express the dendritic cell-specific intercellular adhesion molecule-3-grabbing non-integrin (DC-SIGN) receptor on their surface [13,14], which is a target of mannan-based polymer carriers. Mannans high affinity to DC-SIGN mediated by 3- and 4-OH groups on mannans was previously proved in many other applications than targeting SLNs [15,16,17,18,19].

TAMs and dendritic cells could be targeted not only via their surface DC-SIGN receptor but also via different targeting strategies based on nanotherapeutic, drug delivery, immunotherapeutic or nano-immunotherapeutic approaches [20,21]. More specifically, dendritic cells express not only DC-SIGN receptors, but also toll-like receptors. Some of the immunotherapeutic approaches use toll-like receptors agonists for dendritic cells targeting [22]. Each of the respective approaches possess its own strengths and also limitations. However, we believe that mannan-based copolymers are quite universal and represent a versatile platform.

Mannans are well known for their use in the food industry [23,24], but their myriad biological functions include also storage and cell-wall signaling [25]. Mannan from yeast is formed by D-mannose units connected by α(1-6) bonds (backbone) and α(1-2) and α(1-3) bonds (branches) [26]. In this work, commercially available mannan from *Saccharomyces cerevisiae* was used as a biocompatible platform for mannan conjugate preparation that enables addition of other modalities (imaging probes, etc.). Natural polymers and, especially, polysaccharides such as mannans, provide benefit in that they are biocompatible, biodegradable and widely available within renewable resources [27]. Their sizes can also be easily modified to enable passive accumulation in solid tumors.

The optimal size for passive accumulation of the nanoparticles inside the solid tumors via the enhanced permeability and retention (EPR) effect is approx. up to 200 nm. The EPR effect is caused by a combination of highly permeable newly formed blood vessels and limited lymphatic drainage in tumors [28,29]. Nevertheless, the contribution of the EPR effect to the total accumulation of nanoprobes inside tumors may be affected by physiological features such as tumor intestinal fluid pressure [30,31]. Moreover, there has been a controversy about the EPR effect contribution to the nanoprobes accumulation inside solid tumors. Sindhwani et al. showed that large proportion of the nanoparticles are endocytosed actively into the solid tumors by endothelial cells [32].

Polymer accumulation in tumors via the EPR effect can be influenced by polysaccharides grafting to succinic acid or biocompatible synthetic polymers, a process that decreases the biodegradation rate and enables further chemical modification. This effect has been demonstrated for various types of polysaccharides, such as chitosan and dextrin [33,34]. For our purposes, grafting by poly(2-methyl-2-oxazoline) (POX) was chosen because of its low unspecific organ deposition, low immunogenicity [35,36] and higher stability compared to the polyethylene glycol (PEG) [37]. However, in recent years there is a need for an alternative like POX because of increasing evidence of anti-PEG immunity [38]. There has been reported anti-PEG antibodies not only in pretreated individuals but also in healthy population without previous treatment. The main issue with the anti-PEG antibodies is that they may limit treatment efficacy as well as they enhance adverse effect [39,40] and so the use of the POX seems to be a better grafting modality for nanomedicine field.

In this article, we compared a hybrid copolymer platform based on a biodegradable mannan core grafted with biocompatible hydrophilic POX with a hybrid copolymer without POX to verify the POX effect on accumulation in organs and tumors. Both mannan-based conjugates were grafted also with gadolinium as the MR (magnetic resonance) contrast agent and the fluorescent probe IR800CW for multimodal imaging. POX grafting adjusts the biodegradation rate in dependence on its grafted dose. Both probes were previously tested on healthy animals in pilot experiments [41], with the results indicating favorable biological characteristics for potential use in experimental and clinical medicine.

Here a successful proof of the diagnostic potential of the mannan-based probes for SLNs detection in a relevant tumor animal model is shown together with a detailed toxicity and in vitro/in vivo characterization.

## 2. Results

### 2.1. Chemical Characterizations Are Consistent with Previous Studies

The characteristics of both unmodified and POX modified conjugates were the same as in the previous reported [41]: molecular weight was 52 × 10^3^ g·mol^−1^ for MN-DOTAGd-IR800 (MN) and 71.2 × 10^3^ g·mol^−1^ for MN-pMeOx-DOTAGd-IR800 (MNOX); hydrodynamic radius in PBS was 3.3 nm for MN and 3.6 nm for MNOX. ζ-potential in PBS was −11.5 mV for MN and −6.7 mV for MNOX. Synthesis paths are shown in Appendix A.

### 2.2. MN is Accumulated Inside the Cells in Higher Extent Than MNOX

For determination of subcellular localization of the applied mannan-based carriers 4T1 cells were incubated with a green fluorescent dye for lysosomes (LysoTracker^®^ Green), a blue fluorescent dye for a nucleus and with the fluorescent mannan-based carrier bearing IR800CW as a red fluorescent dye. The confocal microscopy analysis showed that the cells incubated with MNOX had lower fluorescence signal from the red spectrum, this red fluorescent signal was conclusively caused by IR800CW conjugated on mannan-based polymers (Appendix A). The Pearson coefficient, which indicates the co-localization levels of red and green signals (IR800CW and LysoTracker^®^ Green), was also lower in the case of MNOX (0.324 for MNOX and 0.428 for MN), reflecting lower accumulation of the POX-modified polymer in cells compared to the non-modified variant.

### 2.3. MTT Assay Showed That Mannan-Based Polymers Are Not Cytotoxic

Because all carriers intended for future use in drug delivery need to be non-toxic, we performed an MTT cytotoxicity assay. Results from the MTT assay indicated that mannan-based polymers had no negative influence on survival or proliferation of 4T1 cells (Figure 1). No difference was found between control and treated cells. Even the highest concentration used (4.5 mM Gd^3+^) showed no statistically significant cytotoxic effect on cells (all *p*-values ≥ 0.05).

### 2.4. Lymph Nodes on Tumor Site Showed Markedly Higher Fluorescence In Vivo Then Other Lymph Nodes

The analysis of the in vivo distribution, degradation and elimination rate of polymers were performed on tumor-bearing *Balb/cfC3H* mice. Animals were randomized into the three groups (MN, MNOX and DOT) after the tumors establishment. Fluorescence signal was measured repeatedly in several time-points after the injection of the mannan-based carriers. Data from these measurements showed the distribution of the mannan-based carriers in lymph nodes after absorption from the i.m. administration.

According to in vivo fluorescence optical imaging of the MN and MNOX groups (Figure 2), the highest signal issued from the inguinal lymph node next to the tumor and injection site (SLN) (Figure 3a, Appendix A). The signal reached its maximum in both groups between the 4 and 24hours after injection of MN or MNOX, one order of magnitude higher in the case of MN-DOTA Gd-IR800 (as with all other in vivo fluorescent signals). The second highest fluorescent signal measured originated from the liver (Figure 3e, Appendix A), peaking at around 24 h after the injection of MN or MNOX and then continuously decreasing. This signal was again higher in the MN group. The DOT group had always a fluorescence signal lower than the background fluorescence signal.

We later detected the signal from the axillary lymph node on the tumor site (Figure 3b, Appendix A), reaching its maximum 48 h after injection of MNOX (Appendix A) and 72 h after injection of MN. The lowest fluorescent signal of all the measured lymph nodes issued from the axillary lymph node at a non-tumor site (Figure 3c, Appendix A). However, the peak fluorescent signal from this lymph node was recorded only 2 h after administration of the contrast agent, a relatively short time span.

### 2.5. Ex Vivo Fluorescence Signal from Internal Organs Decreased Progressively over Time

Due to the optical signal attenuation in the deeper organs (especially spleen and kidneys), the bio-distribution of the probes was assessed more precisely from ex vivo fluorescence signals (Figure 4, Appendix A) Ex vivo fluorescence signals from all organs decreased progressively over time (days 1, 3 and 7), with the exceptions of tumors and axillary and inguinal lymph nodes on tumor sites in the MNOX group. Fluorescence signal from ex vivo organs was higher for all organs from the MN group with two exceptions-tumors on day 3 and axillary lymph nodes on tumor sites on day 7. In these cases, fluorescence signal was higher in the MNOX group than in the MN group. In the MNOX group, we observed lower fluorescence signals, especially from kidneys (Figure 4f), spleens (Figure 4g) and livers (Figure 4h). However, the trend of gradual decrease of fluorescence signal over time was the same as for the MN group. Fluorescent signal from tumors was the highest on day 1 in the MN group and on day 3 in the MNOX group (Figure 4e), pointing to the slowing of biodegradation due to POX conjugation. In the MNOX group, there was a slight delay in the accumulation of fluorescent probes inside tumors.

### 2.6. MRI Results Correlated with Results from In Vivo Fluorescence

As a complementary imaging method providing no attenuation of the signals in dependence to the tissue localization, we used a standard and non-invasive method routinely used both in experimental medicine and clinical practice: ^1^H-MRI. Focusing on three anatomical sites-both inguinal lymph nodes and respective tumors (Figure 5)-MRI confirmed the results obtained from fluorescence imaging, namely preferential polymer accumulation in SLNs, with significantly higher uptake in animals in the MN and MNOX groups compared to the DOT group (Figure 6). This analysis revealed the highest accumulation in both inguinal lymph nodes in the MN group. The highest accumulation of mannan-based polymer carriers was observed 4 h after injection in the MN group and 24 h after injection in the MNOX group, comparable to fluorescence results. Although mannan-based polymer carrier accumulation in tumors was less prominent than in SLNs, accumulation inside tumors peaked approx. one hour after mannan-based polymer carrier injection (Figure 6c).

The imaging efficacy of the lower-molecular-weight MN probe was significantly higher compared to the higher-molecular-weight MNOX probe (modifying the same mannan with polyoxazoline increases the molecular weight of the conjugate). This might indicate the predominance of the targeting effect of the mannan-DC-SIGN receptor interaction compared to the less effective solid tumor-targeting EPR effect, as evidenced by the molecular weight of the conjugates.

### 2.7. Histology Did Not Find Any Pathologies in the Internal Organs after the Mannan Polymers Application

Histological analysis was performed in order to exclude pathological changes in the examined organs after the exposure to the mannan-based polymer carriers and to enable detailed monitoring of the induced tumors. No macroscopic or light-microscopic pathologies were observed for any of the investigated organs (Figure 7). Light microscopic examination revealed the preservation of tissue architecture, with no signs of dystrophic/degenerative change, necroinflammmatory activity or fibrosis. No unexpected neoplastic processes were discernible in the organs examined (Figure 7(A1–A3),(B1–B3),(C1–C3)). Histological examination of the induced tumors revealed uncircumscribed masses composed of poorly differentiated, frankly atypical neoplastic cells with hyperchromatic nuclei, markedly increased nucleo-cytoplasmic ratios, high mitotic activity, and atypical mitoses. Areas of coagulative necrosis were identified within neoplastic tissue.

## 3. Discussion

In this study, we tested a novel versatile platform based on mannan polymers that is intended for detection of SLNs infiltrated by metastasis and for tumor theranostics. We chose the 4T1 model, which is characterized by SLN infiltrated by metastatic cancer cells, a very common occurrence in breast cancer. Mannan-based polymers were modified with imaging probes for subsequent tracking. From a chemical point of view, the mannan-based probes were found to be stable, providing signals sufficient for both MRI and optical fluorescence imaging.

Confocal microscopy images showed lower fluorescent signals originating from MNOX carriers compared to carriers without polyoxazoline. MN and MNOX both co-localized with lysosomes to a similar extent; however, the Pearson coefficient of co-localization was lower in the case of the POX-modified conjugate due to lower intracellular uptake. The probe modified with POX seemed to exhibit lower endocytosis efficacy than MN without POX. This may have been caused partly by POX functionalization, which can hide mannan-based polymer carriers and thus preventing them from actively interacting with DC-SIGN receptors. POX functionalization can also affect the endocytosis rate. Previous studies focusing on other types of chemical functionalization (dextrin succinylation [33,34] or PEGylation [42]) have documented similar results.

In vivo multimodal imaging of mice with syngeneic tumors confirmed in vitro results. The overall higher in vivo fluorescence signals of MN in the lymph nodes and liver most likely indicates higher accumulation of the polymer without POX. This tallies with our hypothesis and with previous studies which have shown that POX prolongs polymer circulation time, decreases accumulation in internal organs, and renders polymers less visible to the immune system [33,34]. Lower accumulation in organs such as the liver, kidneys and spleen, is beneficial because it diminishes the unwanted effects in off-target organs. On the other hand, a disadvantage of the modification is that the accumulation is also reduced in tumor and lymph nodes. Assisted by passive targeting via the EPR effect, MN offers stronger primary active targeting than MNOX because of its high affinity with DC-SIGN receptors. This is of no small importance given that DC-SIGN receptors are present on macrophages, especially those inside SLNs, the most common site of primary metastasis. Active targeting of this problematic site is therefore recommended, even at early stages of tumorigenesis. Our mannan-based polymer with POX seemed to have less availability of mannose for DC-SIGN receptors and slightly lower content of IR800CW. Further analysis is required to precisely equilibrate the dose of POX in this mannan-based conjugate to maximize its potential benefits, namely lower accumulation in organs, but still preserve the strong tumor- and lymph node-targeting properties.

Although polymers were accumulated mostly in SLNs, we also observed accumulation in other lymph nodes. The extent of accumulation in distant lymph nodes enabled us to monitor the spreading of our probe through lymphatic drainage. In accordance with fluorescence imaging, MRI measured direct signals from SLNs without signal contamination from the injection site and confirmed preferential uptake of mannan-based conjugates in lymph nodes compared to that offered by the commercially available MR contrast agent, gadoterate meglumine.

Ex vivo fluorescence signals from organs (liver, spleen, kidneys) decreased over time in the case of both mannan-based conjugates, a trend that indicates the gradual degradation and elimination of our polymer carriers. Degradation and elimination were slower for the POX probe, a finding that corresponds with other in vivo studies of natural polysaccharide-based conjugates [33,34]. Gradual biodegradation and elimination should prove beneficial characteristics for future clinical applications [43]. However, fluorescence signal activity in tumor and lymph nodes proved different. The decrease in fluorescence signal was not continuous for the MNOX group in the case of SLNs, ALNs on tumor sites, or tumors. Additionally, in both in vivo and ex vivo applications, the fluorescence signal from MN probes were higher with two exceptions: axillary lymph nodes on tumor sites on day 7 and tumors on day 3, where MNOX exhibited a higher fluorescence signal. These results may be due to the effect of POX, namely slower cell uptake, diminished biodegradation and prolonged circulation time.

Importantly, our results confirm the biocompatibility and non-toxicity of mannan-based carriers. The MTT assay results conclusively support our hypothesis that mannan-based polymers are non-toxic even when tested at maximum concentrations. Histological examination confirmed our mannan-based polymer carriers had no adverse effects on internal organs. As previously shown, Gd^3+^ can have a negative effect when cleaved from its structure [44,45,46,47]; however, as Gd^3+^ was chelated in mannan-based conjugate, its toxic effect was excluded. Furthermore, Gd^3+^ chelates are approved and routinely used for medical purposes, which suggests that our carriers could be easily implemented in clinical practice.

The polymers described here might not only serve as a diagnostic aid for tumors and metastasis but also as a beneficial therapeutic option. In future, drugs might be chemically incorporated into these polymers to serve as efficient drug delivery systems via local drug release in response to various stimuli (e.g., pH).

## 4. Materials and Methods

### 4.1. Chemistry

Mannan-based conjugates were prepared as described in [41]. Briefly, mannan was modified in two different ways to obtain conjugate bearing a fluorescent label and a probe for MRI.

The first approach was focused on the synthesis of polysaccharide-based conjugate without polyoxazoline in the structure (Appendix A). The modification procedure was started by allylation of commercial mannan to obtain allyl groups-containing derivative. It was further reacted with cysteamine via thiol-click reaction introducing primary amino groups. This primary amino group-containing mannan was then conjugated with N-hydroxysuccinimide (NHS) esters of infra-red dye (IR800CW NHS-ester) and 1,4,7,10-tetraazacyclododecane-1,4,7,10-tetraacetic acid (DOTA NHS-ester). Finally, the obtained product reacted with gadolinium (III) chloride to chelate Gd^3+^, resulting in mannan-based conjugate with the fluorescence and MR imaging labels, denoted as MN-DOTAGd-IR800 (in short version-MN).

Synthetic approach for preparation of mannan-based conjugate with grafted polyoxazoline chains was analogous to MN-DOTAGd-IR800 (Appendix A). After dissolution in anhydrous DMSO, mannan sodium alkoxide was reacted with living poly(2-methyl-2-oxazoline) chains, obtained by ring-opening cationic polymerization. As allyl bromide was used for the polymerization of the polyoxazoline, the grafts contained terminal allyl groups which were then modified in the same way as described above for MN-DOTAGd-IR800. The polyoxazoline-containing conjugate was denoted as MN-pMeOx-DOTAGd-IR800 (in short version-MNOX).

### 4.2. Cell Line

For all in vitro and in vivo experiments, 4T1 cells (ATCC^®^ CRL-2539™, Prague, Czech Republic) were used. The cells were incubated under standard conditions (37 °C, 5% CO_2_) in Roswell Park Memorial Institute (RPMI) 1640 incubation medium without phenol red supplemented with fetal bovine serum, L-glutamine and penicillin/streptomycin. RPMI 1640 incubation medium was purchased from Gibco^®^ by LifeTechnologies™ (Waltham, MA, USA).

RPMI 1640 incubation medium without phenol red was chosen in order to minimize background during fluorescent microscopy measurement. Fetal bovine serum that was added to RPMI 1640 media to final concentration 10% was purchased from Gibco^®^ by LifeTechnologies™. L-Glutamine (stock solution 200 mM) and penicillin/streptomycin (stock solution containing 10,000 units of penicillin and 10 mg of streptomycin per 1 mL) were added to the RPMI 1640 media at a final concentration of 5%. Both L-glutamine and penicillin/streptomycin were purchased from Sigma-Aldrich Ltd. (Prague, Czech Republic).

The RPMI 1640 incubation medium without phenol red supplemented with fetal bovine serum, L-glutamine and penicillin/streptomycin was used as a medium for all experiments with 4T1 cell line (confocal microscopy, MTT cytotoxicity assay).

### 4.3. Confocal Microscopy

The 4T1 cells (0.1 × 10^6^/mL) were plated in an 8-well Nunc™ Lab-Tek™ II Chambered Coverglass dish (Thermo Scientific™, Waltham, MA, USA) with a No. 1.5 borosilicate glass bottom. In their exponential phase of growth, cells were incubated for 24 h in media with a 4.5 mM Gd^3+^ con-centration of mannan-based polymers. After the incubation period, cells were washed twice with Hank’s balanced salt solution (HBSS, Biosera, Nuaille, France), with fluorescent dyes added in concentrations according to the producer’s manual (60–70 nM for LysoTracker^®^ Green and 1 μg/mL for Hoechst 33342). Incubation times were 60 min for LysoTracker^®^ Green and 20 min for Hoechst 33,342 [48,49]. All fluorescent dyes were purchased from Invitrogen™ by Life-Technologies (Prague, Czech Republic). After incubation, the cells were washed twice with HBSS followed by the addition of RPMI 1640 medium without phenol red. Cells were then measured under a TCS SP8 STED 3× microscope (Leica, Chicago, IL, USA; objective: HC PL APO CS2 100×/1.40 OIL). Images were displayed with automatically enhanced contrast and adjusted for brightness using ImageJ (version 1.46r, National Institutes of Health, Bethesda, MD, USA).

### 4.4. MTT Cytotoxicity Assay

The MTT (3-(4,5-dimethylthiazol-2-yl)-2,5-diphenyltetrazolium bromide) assay was performed according to a standard protocol [50]. For the cytotoxicity test, the 4T1 cell line (incubated in RPMI 1640 medium without phenol red supplemented with fetal bovine serum, L-glutamine and penicillin/streptomycin) was used at a concentration of 0.01 × 10^6^/mL. The highest concentration of the tested sub-stance (4.5 mM of Gd^3+^) was used to reflect the optimal concentration for all other in vitro experiments. Fivefold serial dilution of the initial concentration was used to prepare other samples. After 24 h of incubation with the mannan-based polymers (MN or MNOX), cells were washed with HBSS before adding 200 μL of fresh media, and then incubated for another 5 days. The medium was removed followed by the addition of the MTT solution (5 mg/mL in RPMI 1640, 250 μL per well) for 6 h. The MTT solution was then replaced with dimethyl sulfoxide (DMSO, 200 μL per well) and glycine buffer (30 μL per well). Absorbance at 570 nm was immediately measured on the Multi-Mode Reader (Synergy™ 2, BioTek^®^Instruments, Inc., Burlington, VT, USA), with tests performed in tetraplicate. Both MTT and DMSO were purchased from Sigma-Aldrich, Ltd.

### 4.5. Animal Model

For all in vivo measurements, female 5-week-old *Balb/cfC3H* mice were used. With this animal strain, syngeneic tumors can be induced by injecting 4T1 cells. The animals (purchased from Velaz Ltd., Prague, Czech Republic) were kept under a standard day/night cycle (12/12 h) and given free access to food and water. All protocols were approved by the Ethics Committee of the Institute for Clinical and Experimental Medicine, with all experiments carried out in accordance with European Union Council Directive 2010/63/EU.

### 4.6. Tumor Induction

4T1 cells were chosen for tumor induction. The cell line originates from the mammary gland of *Mus musculus* (*Balb/cfC3H* strain), representing an animal stage IV human breast cancer. These cells allow syngeneic tumor induction in *Balb/cfC3H* mice and can form metastases from primary tumors. Additionally, tumors formed from 4T1 cells have homogeneous regions, which are beneficial for imaging and subsequent quantification.

Tumors were induced in *Balb/cfC3H* mice by an injection of 0.30 ± 0.05 × 10^6^ 4T1 cells suspended in 50 μL PBS into the right abdominal mammary gland. Cells were harvested under standard conditions (37 °C, % CO2). On the day of the injection, cells were trypsinized, centrifuged, counted, diluted in PBS at the desired concentration, and then injected into the anesthetized animals, which were kept under inhalation anesthesia using isoflurane (5% for induction, 2% during the surgery).

When tumors reached at least 2 mm in diameter (assessed by MRI), the animals were divided into three groups: MNOX group-i.m. administration with MN-pMeOx-DOTAGd-IR800 (n = 9); MN group-i.m. administration with MN-DOTAGd-IR800 (n = 9); DOT control group-i.m. administration with gadoterate meglumine, a clinically approved contrast agent (n = 6). Gadoterate meglumine was chosen as a control due to its common use in clinical practice thus the signal mannan-based conjugates could be directly compared to widely use contrast agent.

The probes (50 µl dose) were injected into the right tight muscle at an 18 mM Gd^3+^ concentration per ml (0.05361 mg IR800CW per ml in case of MN or 0.04932 mg IR800CW per ml in case of MNOX). 18 mM Gd^3+^ concentration gives signal that is strong enough for measurement with 4.7 T MR scanner and subsequent analysis.

### 4.7. Fluorescence Imaging

IVIS^®^ Lumina XR optical imager (PerkinElmer Inc., Waltham, MA, USA) (excitation filter: 745 nm, emission filter: 810–875 nm) was used for in vivo experiments. The animals were scanned (exposure time: 60 s) at several time points: before, immediately, and then 2, 4, 6, 24, 48, 72 and 168 h after the injection. After fluorescence imaging, animals were measured by MRI (except at the 48-h time point). Two animals from each group were sacrificed on the 1st and 3rd day and five animals on the 7th day after the polymer injection. Fluorescent signals from organs (liver, kidneys, spleen, lymph nodes and tumors) were also measured to determine level of fluorescence in these sites.

### 4.8. Magnetic Resonance Imaging

MRI examination was carried out on a 4.7 T MR scanner (Bruker BioSpin, Ettlingen, Germany) using a homemade surface coil. T1-weighted axial and coronal MR images of mouse calf muscles and lymph nodes were acquired via standard two-dimensional rapid acquisition and the relaxation enhancement (RARE) multispin echo sequence using the following parameters: repetition time TR = 339 ms, effective echo time TE = 12 ms, turbo factor 2, spatial resolution 0.16 × 0.16 × 0.70 mm^3^, scan time 8 min 40 s. Data were analyzed and presented as percentages of the signal- (from the appropriate lymph node or tumor mass) to-noise ratio (SNR), with values measured before probe application at 100%.

### 4.9. Histology

Livers, kidneys, SLNs and tumors were analyzed histologically. Two mice from each group (MN, MNOX and DOT) were randomly chosen for analysis. Mice were sacrificed by anesthesia overdose 7 days after polymer application. After a thorough macroscopic inspection, internal organs (kidneys, livers, lymph nodes) as well as tumor tissue were fixed in 4% formaldehyde and routinely processed for histological examination. Sections (4µm) were stained with hematoxylin and eosin (HE) and the Verhoeff-van Gieson protocol to highlight collagen and elastic fibers.

### 4.10. Statistical Methods

The R 3.6.2 a language and environment for statistical computing (R Foundation for Statistical Computing, Vienna, Austria) was used for statistical analysis of MTT assay data using linear mixed-effects model (lme4 package). For fluorescence in vivo imaging and MRI the multiple t-test in GraphPad Prism 8 (GraphPad Software, Inc., San Diego, CA, USA) was used.

## 5. Conclusions

We showed in this article the diagnostic potential of the mannan-based probes for SLNs in a relevant fully immunocompetent tumor animal model. The conjugates were accumulated in vitro inside 4T1 cells and in vivo after intramuscular administration in the lymph nodes and internal organs of mice. No toxic effects were observed, the conjugates proved highly biocompatible. The probes were preferentially accumulated in SLNs and tumors in the same animal model. Fluorescence imaging confirmed the biodegradability of the probes (with or without POX modification). Considering the promising potential of polymer conjugates as a precise and efficient theranostic multimodal imaging modality, the next step will be to test their application in various anti-cancer drug systems. The binding with various anti-cancer drugs targeted to specific cancer subtype could overcome the limitation of the mannan-based probes (targeting to DC-SIGN only), increase their efficiency and conjugated anti-cancer drugs could benefit from the mannan-based probes versatility, targeting via EPR effect and possibility of local anti-cancer drug release in response to various stimuli (which would depend on the type of conjugation). Therefore, these combinations could form a very promising various drug delivery systems based on the mannan copolymer platform which have been presented and described in this article.

## Figures and Tables

**Figure 1 molecules-26-00146-f001:**
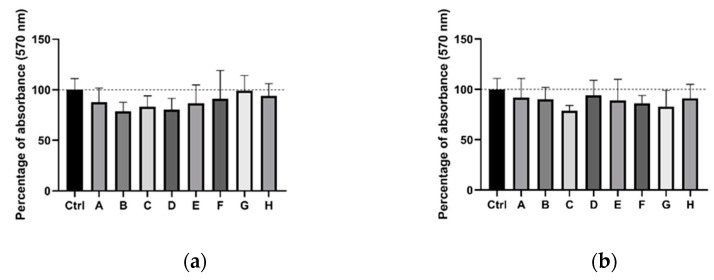
MTT assay results for 4T1 cells after 24 h of incubation with: (**a**) MN and; (**b**) MNOX. A-H represent different concentrations of MN and MNOX. Column A represents the highest concentration (4.5 mM Gd^3+^) followed by fivefold serial dilution until the lowest concentration, represented by the H column. Data are displayed as a percentage of mean (±SD) absorbance (n = 4), 100% is the mean signal of untreated cells (controls-Ctrl).

**Figure 2 molecules-26-00146-f002:**
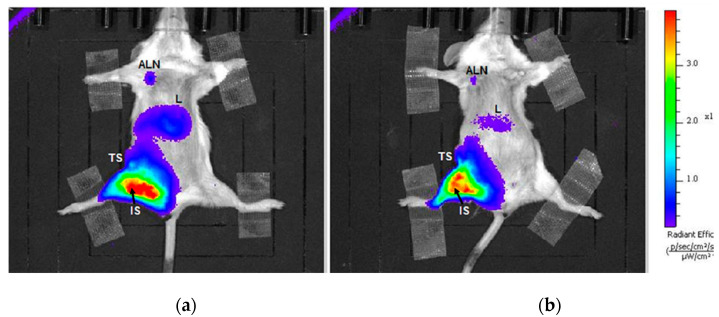
Representative images of in vivo fluorescence from (**a**) MN and (**b**) MNOX group. Fluorescent signals from axillary lymph nodes (ALN), livers (L), tumor sites (TS) and injection sites (IS). Images show mice three days after intramuscular injection of (**a**) MN or (**b**) MNOX.

**Figure 3 molecules-26-00146-f003:**
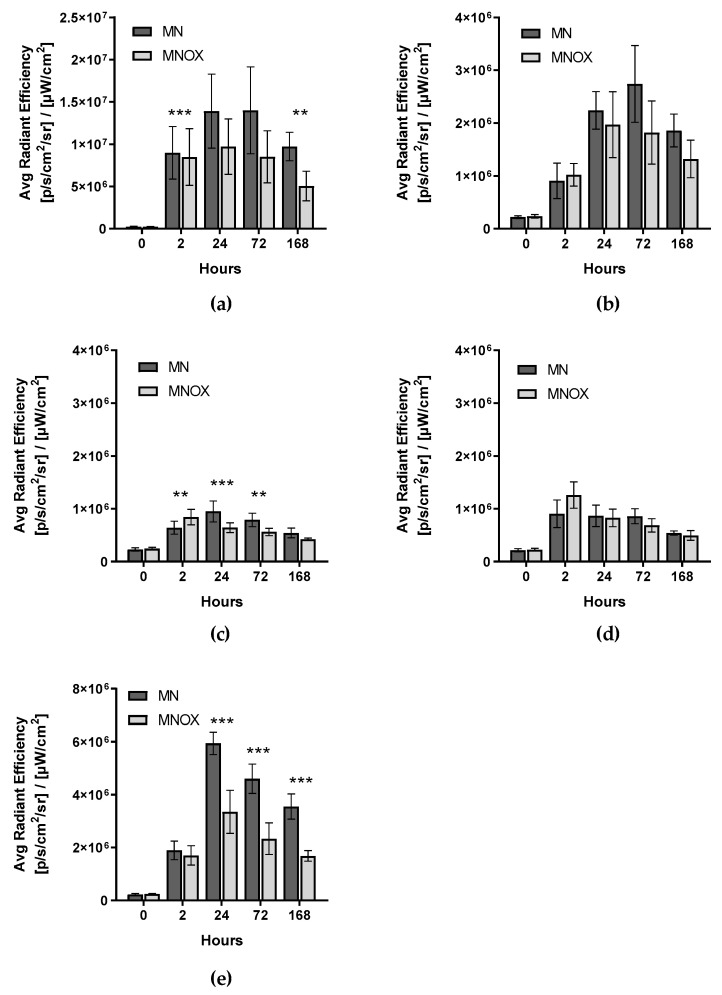
Quantification of fluorescence signals in different time intervals in vivo. Signals were quantified from (**a**) SLNs (inguinal lymph nodes on tumor site), (**b**) axillary lymph nodes on tumor sites, (**c**) axillary lymph nodes on non-tumor sites, (**d**) inguinal lymph nodes on non-tumor sites and (**e**) livers. The multiple one-tailed t-test was used for statistical evaluation (n = 9 for MN and n = 9 for MNOX). The fluorescent signal is represented as the average radiant efficiency (mean ± SD), *p*-values: *** *p* < 0.001, ** *p* < 0.01, * *p* < 0.05.

**Figure 4 molecules-26-00146-f004:**
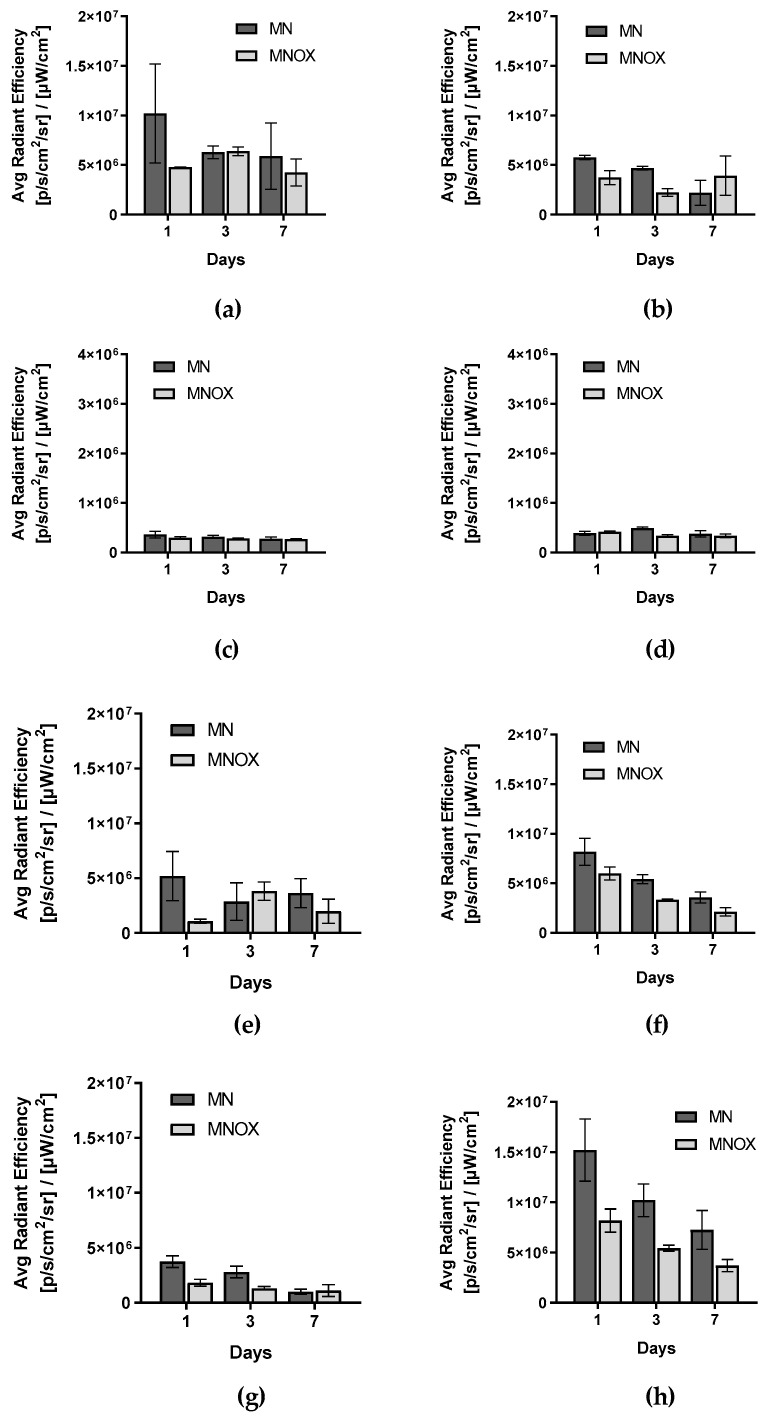
Figure 5. Quantification of ex vivo fluorescence signals. The fluorescence signal was quantified for (**a**) SLNs, (**b**) axillary lymph nodes on tumor sites, (**c**) axillary lymph nodes on non-tumor sites, (**d**) inguinal lymph nodes on non-tumor sites, (**e**) tumors, (**f**) kidneys, (**g**) spleens and (**h**) livers. Fluorescent signal is represented as the average radiant efficiency (mean ± SD). The multiple one-tailed t-test was used for statistical evaluation (n = 9 for MN and n = 9 for MNOX).

**Figure 5 molecules-26-00146-f005:**
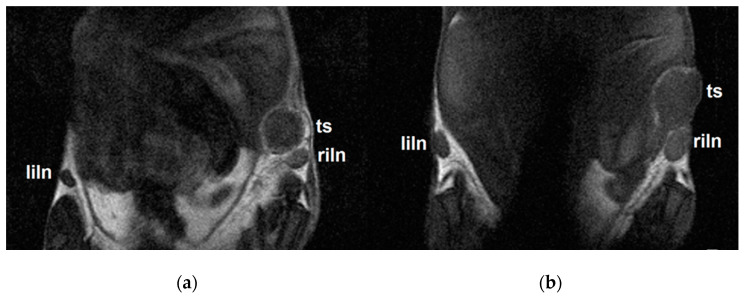
Representative coronal MR images measured three days after the injection of (**a**) MN or (**b**) MNOX into the right thigh muscle. Left inguinal lymph nodes (liln), tumors (ts) and right inguinal lymph nodes (riln) represent sites of polymer accumulation.

**Figure 6 molecules-26-00146-f006:**
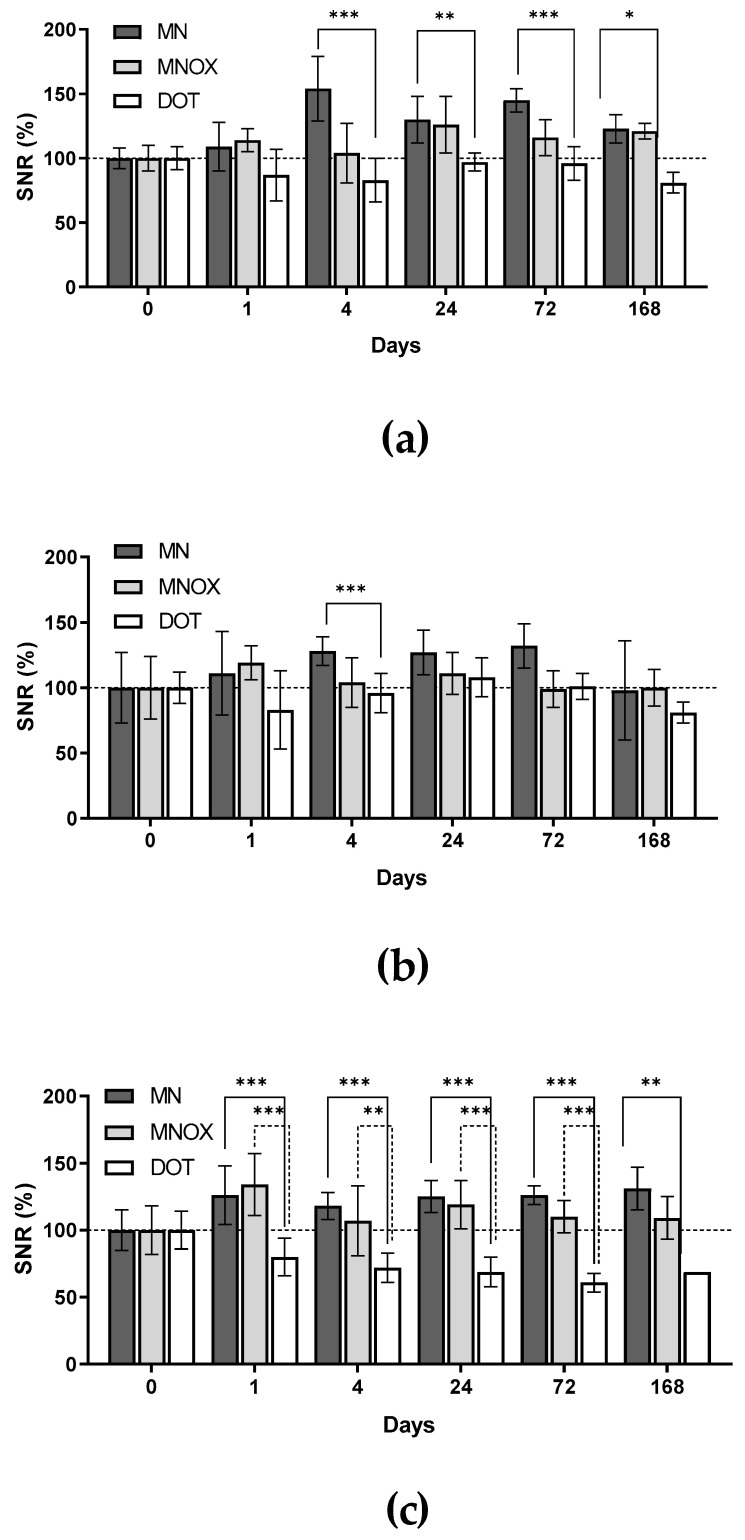
Quantification of MR measurements. Results are presented as the SNR percentage mean ± SD. Measurements performed before application of contrast agents are deemed 100%. Graphs show results from (**a**) inguinal lymph nodes on non-tumor sites, (**b**) inguinal lymph nodes on tumor sites (SLNs) and (**c**) tumors. The multiple one-tailed t-test was used for statistical evaluation (n = 9 for MN group, n = 9 for MNOX group and n = 6 for DOT group), *p*-values: *** *p* < 001, ** *p* < 0.01, * *p* < 0.05.

**Figure 7 molecules-26-00146-f007:**
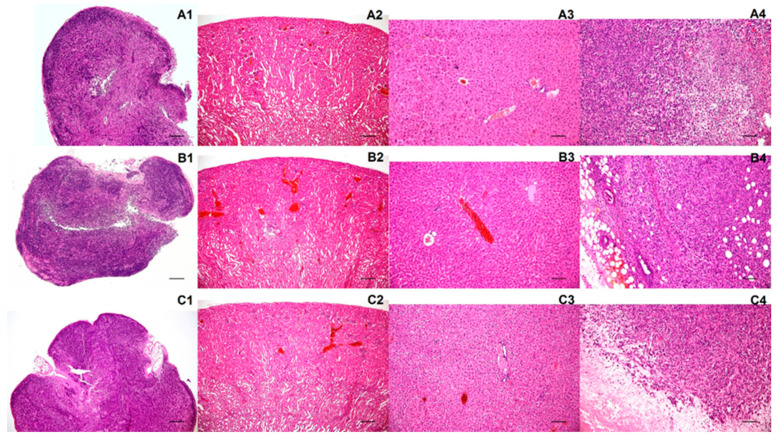
Representative histology images. Images represent (**A1**–**A4**) organs from MN-treated animals; (**B1**–**B4**) organs from MNOX-treated animals; (**C1**–**C4**) organs from animals treated with gadoterate meglumine. The following organs were analyzed: lymph nodes (**A1**,**B1**,**C1**), kidneys (**A2**,**B2**,**C2**), livers (**A3**,**B3**,**C3**) and tumors (**A4**,**B4**,**C4**). Hematoxylin-eosin staining; scale bars for **A1**–**A3**, **B1**–**B3**, **C1**–**C3** represent 100× magnification, with those for **A4**, **B4** and **C4** representing 400× magnification.

## Data Availability

The data presented in this study are available on request from the corresponding author.

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
