# Peer review of "Mannan-Based Nanodiagnostic Agents for Targeting Sentinel Lymph Nodes and Tumors"

_molecules, 2020, doi:10.3390/molecules26010146_

Round 1

Reviewer 1 Report

The article presented by Jirátová and collaborators is quite interesting covering a relevant topic in diagnostic agents. However it needs some improvments before its publication:

major

chemical characterization in my opinion is not completely accomplished. Possibly, NMR characterization (1H and 13C) could add a reliability regarding the molecules obtained by the synthesis

moderate english revision is required to avoid some mistakes about the grammar

introduction, despite a clear attempt to provide the state-of-the-art, in my opinion a short paragraph regarding compounds able to target the selected cellular systems should be clearly reported including a table with drawbacks and strength of the actual developed systems.

in order to provide a better reading of the manuscript I suggest to move materials and methods in the section 2, results section 3, discussion section 4 and conclusion section 5.

In the conclusion, the authors should provide the limitations of these compounds and the suggestions to improve their efficiency.

minor

acronyms in the abstract should be defined such as DC-SIGN

names of organisms such as Saccharomyces cerevisiae should be reported in italic

in the conclusion section authors wrote sentinel lymph nodes that should be replaced by the acronym previously introduced (SLNs). please check acronyms and their definition within the text.

Author Response

The article presented by Jirátová and collaborators is quite interesting covering a relevant topic in diagnostic agents. However it needs some improvments before its publication:

Comments and Suggestions for Authors:

major

  • chemical characterization in my opinion is not completely accomplished. Possibly, NMR characterization (1H and 13C) could add a reliability regarding the molecules obtained by the synthesis

Dear reviewer, thank you for your comment and the suggestion. However, we were more focused on the biological characteristics of the mannan-based molecules and their capability to be visualized in an animal model. More detailed chemical characterization (including 1H relaxometry and NMR characterization) could be found in this article: Rabyk M et al. Mannan-based conjugates as a multimodal imaging platform for lymph nodes. J Mater Chem B. 2018 May 7;6(17):2584-2596. This article is also included in the references of the manuscript.

  • moderate english revision is required to avoid some mistakes about the grammar

We went through the article and correct the English grammar. Unfortunately, none of the co-authors is native English speaker. Therefore, the manuscript was sent for English Proof reading before submitting. However, after that some little changes were made but we believe it is not cardinal for paper clarity.

  • introduction, despite a clear attempt to provide the state-of-the-art, in my opinion a short paragraph regarding compounds able to target the selected cellular systems should be clearly reported including a table with drawbacks and strength of the actual developed systems.

We think that this is a great suggestion for extension of the scope of the article. However, we believe that this topic is far too complex to be concisely and clearly described in just one paragraph. Also, this topic could be described with many different points of view (it means more chemical-based approach, immuno oncology based approach etc.), therefore we added short paragraph with some hints to this complex issue and with reference to the relevant reviews where readers could find more complex description, please see page 2, lines 57-63.

  • in order to provide a better reading of the manuscript I suggest to move materials and methods in the section 2, results section 3, discussion section 4 and conclusion section 5.

Thank you for your suggestion, we thought about it previously. However, we have followed the structure that is recommended for the Molecules journal (actually, we have used the template from their webpage). If the editor office would request these changes, we will change the order of the sections according to your suggestion.

  • In the conclusion, the authors should provide the limitations of these compounds and the suggestions to improve their efficiency.

Thank you for this comment, we have added a short description of possible beneficial combinations for overcoming limitations of the mannan-based copolymers and improving their overall efficiency, please see page 14, lines 426-432. Limitations of the presented mannan-based copolymers are also discussed in section 3 (for example page 10, line 254-256).

minor

  • acronyms in the abstract should be defined such as DC-SIGN

Acronyms in the abstract were checked and DC-SIGN acronym was defined, please see page 1, line 17-18.

  • names of organisms such as Saccharomyces cerevisiae should be reported in italic

We would like to thank you for noticing this mistake, we have gone through the article and changed all the name of all organism to italic font (please see page 2, line 67 with the correction) as well as in vivo, ex vivo and in vitro terms. All changes can be seen in track version of the revised manuscript.

  • in the conclusion section authors wrote sentinel lymph nodes that should be replaced by the acronym previously introduced (SLNs). please check acronyms and their definition within the text.

The sentinel lymph nodes term was changed to SLNs in the conclusion, please see page 14, line 422 Usage of all other acronyms and their definitions were checked through the article and adapted, if needed.

Reviewer 2 Report

In their submitted work, Jiratova and colleagues developed Mannan-based particles that target cancer tumours. They tested the cytotoxicity of their probe and investigated its usage in a mouse model. Overall, I liked the story and have only a few points to comment on: 

Figure 1: Please indicate how many repeats of the experiment was performed.

Line 133: in vivo should be in italic. (same for other uses of in vivo and in vitro in the text)

The description of the cell culture in the method section is quite confusing. Would worth rewriting it? At the moment, it is unclear what is in the media, what was diluted and why. 

Author Response

Reviewer #2

In their submitted work, Jiratova and colleagues developed Mannan-based particles that target cancer tumours. They tested the cytotoxicity of their probe and investigated its usage in a mouse model. Overall, I liked the story and have only a few points to comment on: 

Comments and Suggestions for Authors:

  • Figure 1: Please indicate how many repeats of the experiment was performed.

There were 4 repeats (n=4). This information was added to the figure description, please see page 4, line 132.

  • Line 133: in vivo should be in italic. (same for other uses of in vivo and in vitro in the text)

Many thanks for noticing, unfortunately we have missed this in the original article. All mentioned terms (in vitro, in vivo as well as ex vivo) were changed in the revised manuscript and now are in italic font. All changes are indicated in track version of the revised article.

  • The description of the cell culture in the method section is quite confusing. Would worth rewriting it? At the moment, it is unclear what is in the media, what was diluted and why. 

Thank you for your suggestion, we have edited the section 4.2 and 4.4, please see page 12, lines 319-331 and 347-349. We hope that now it is easier to understand what was included in the incubation media formulation and why was the specified medium chosen.